# The Imaging of Primary Fallopian Tube Carcinoma: A Literature Review

**DOI:** 10.3390/cancers17182985

**Published:** 2025-09-12

**Authors:** Giulia Iacobellis, Alessia Leggio, Cecilia Salzillo, Amalia Imparato, Andrea Marzullo

**Affiliations:** 1Radiology Unit, Department of Clinical and Experimental Medicine, University of Foggia, 71122 Foggia, Italy; 2Legal Medicine Unit, Department of Interdisciplinary Medicine, University of Bari “Aldo Moro”, 70121 Bari, Italy; a.leggio@eufor.eu; 3Pathology Unit, Department of Precision and Regenerative Medicine and Ionian Area, University of Bari “Aldo Moro”, 70121 Bari, Italy; cecilia.salzillo@uniba.it; 4Department of Experimental Medicine, PhD Course in Public Health, University of Campania “Luigi Vanvitelli”, 80138 Naples, Italy; 5Obstetrics and Gynecology Unit, Department of Woman, Child and General and Specialized Surgery, University of Campania “Luigi Vanvitelli”, 80138 Naples, Italy; amalia.imparato@studenti.unicampania.it

**Keywords:** primary Fallopian tube carcinoma, ovarian cancer mimic, diagnostic imaging, gynecological malignancy, US, MRI, PET-CT

## Abstract

**Simple Summary:**

Primary Fallopian tube carcinomas (PFTCs) are rare gynecological malignancies that are notoriously challenging to diagnose at an early stage due to their nonspecific symptoms and their resemblance to ovarian cancer. This review consolidates current knowledge on their clinical presentation and diagnostic approaches, with specific emphasis on strategies of imaging for the early identification of the disease. By synthesizing existing evidence, we highlight how enhanced diagnostic awareness and refined imaging protocols can improve early detection. These findings aim to improve the recognition of early-stage Fallopian tube malignancies, thereby advancing patient care and guiding future research in gynecologic oncology.

**Abstract:**

Primary Fallopian tube carcinomas (PFTCs) are rare malignancies that are often misclassified as ovarian cancers due to overlapping clinical and pathological features. This frequent misdiagnosis contributes to the under-recognition of PFTCs, which account for a larger proportion of pelvic malignancies than historically reported. The central aim of this literature review is to highlight the critical importance of and methods for achieving an early diagnosis of Fallopian tube cancer, to improve patient outcomes. We classify benign and malignant fallopian tube neoplasms and evaluate the essential role of clinical evaluation and advanced imaging techniques, considering especially ultrasound, MRI, and PET-CT, in achieving an accurate and timely diagnosis. While histopathology remains the gold standard, imaging is pivotal for differentiating benign from malignant tubal lesions. This review details clinical manifestations, diagnostic pitfalls, and the necessity of a multidisciplinary approach to management. We conclude that advancing early detection through refined diagnostic criteria is essential to guiding effective, patient-specific therapeutic interventions.

## 1. Introduction

Primary Fallopian tube carcinomas (PFTCs) represent a rare (0.3–1.1% of gynecologic cancers) but clinically significant diagnostic challenge [1]. PFTCs demonstrate peak incidence among perimenopausal women, with many cases occurring between the fourth and sixth decades of life. Most cases are discovered incidentally during surgery or on final pathology due to their mimicry of ovarian carcinomas and the predominance of metastatic tubal tumors [2]. While classically presenting with vaginal bleeding, pelvic pain, and watery discharge (Latzko’s triad), these nonspecific symptoms rarely permit preoperative diagnosis [3]. The diagnostic complexity stems from three key factors: shared molecular origins with high-grade serous ovarian cancers (both often arising from tubal fimbriae), similar treatment protocols to ovarian cancer, and frequent misclassification in imaging studies [4,5]. The World Health Organization (WHO) spearheaded a comprehensive revision of the classification system for ovarian, Fallopian tube, and peritoneal cancers in 2014, with parallel updates to the FIGO surgical staging system for these malignancies [5]. These coordinated revisions reflect an evolving understanding of disease pathogenesis, aiming to improve both biological characterization and clinical management. The FIGO staging system encompasses all histological subtypes, including epithelial, stromal, and germ cell tumors, with cases of indeterminate primary origin classified as “undesignated” [6].

However, advances over the past two decades have demonstrated that most high-grade serous carcinomas (HGSCs), traditionally attributed to the ovary or peritoneum, originate from the tubal fimbria. This has led to a paradigm shift whereby tubo-ovarian and peritoneal serous carcinomas should be considered as a single biological entity, part of a continuum. In this context, the close comparison between ovarian and tubal cancer is no longer clinically useful, except in the very rare cases in which tubal cancer is diagnosed at stage I [5,7].

It is precisely these exceptional cases that represent the true diagnostic challenge and opportunity. Indeed, early detection of tubal cancer, often in patients with BRCA mutations undergoing prophylactic surgery, can allow for curative treatment before ovarian or peritoneal dissemination. Symptoms such as vaginal bleeding or discharge associated with a dilated tube, in the presence of a negative cervical and endometrial evaluation, should raise the possibility of early tubal carcinoma.

Lymphatic drainage follows predictable pathways through utero-ovarian, infundibulopelvic, and round ligament routes to regional nodes (external/internal iliac, hypogastric, lateral sacral, para-aortic, and occasionally inguinal) [8]. Peritoneal surfaces may drain via diaphragmatic lymphatics to supradiaphragmatic venous circulation. The peritoneum, including omentum and visceral surfaces, represents the most common site of spread, with characteristic diaphragmatic and hepatic involvement. While peritoneal/pleural involvement occurs frequently, extraperitoneal/extrapleural metastases remain uncommon. Notably, primary peritoneal malignancies demonstrate identical dissemination patterns and may secondarily involve adnexal structures [8].

This review critically evaluates the pivotal role of advanced imaging techniques, including high-resolution MRI with quantitative and qualitative feature analysis, in the preoperative differentiation. of benign from malignant tubal lesions.

By evaluating the diagnostic accuracy of these imaging biomarkers, the review addresses a key clinical challenge improving preoperative differentiation to guide tailored surgical and therapeutic strategies, such as lymphadenectomy or surgical approach, while underscoring the limitations and future directions of current imaging paradigms.

## 2. Materials and Methods

This literature review was conducted following a systematic approach to identify, select, and analyze the most relevant studies on primary Fallopian tube carcinomas (PFTCs), with particular focus on imaging diagnosis. A comprehensive search strategy was implemented across major biomedical databases including PubMed/MEDLINE, Scopus, Web of Science, and Embase. The search employed combinations of key terms such as “primary Fallopian tube carcinoma,” “Fallopian tube cancer,” “ovarian cancer mimic,” “tubo-ovarian malignancy,” “imaging diagnosis,” “MRI,” “ultrasound,” and “PET/CT.” The search was limited to English-language publications without temporal restrictions to ensure a complete overview of diagnostic evolution.

Study selection criteria prioritized original research articles, meta-analyses, and systematic reviews with histological confirmation of PFTCs and evaluation of imaging modality accuracy (including MRI, ultrasound, and PET/CT). Studies were required to report either sensitivity/specificity data or distinctive radiological features. As a literature review, this study did not require specific ethical approvals, though all cited studies declared compliance with original ethical regulations. Extracted data are available upon request, with all selected studies being publicly accessible through the referenced databases.

## 3. Clinical Features

Primary Fallopian tube carcinoma represents a diagnostic challenge due to its nonspecific clinical presentation and incompletely understood etiology. While hormonal, reproductive, and genetic factors, along with chronic pelvic inflammation, have been implicated in its pathogenesis, the precise mechanisms remain elusive [9]. The classic Latzko’s triad characterized by colicky pelvic pain, adnexal mass, and intermittent serosanguinous vaginal discharge—presents in only 15% of cases, with the even rarer hydrops tubae profluens phenomenon (vaginal discharge-associated symptom relief) occurring in merely 5% of patients [9].

Most cases occur in perimenopausal women (median age 55 years), typically manifesting with vague symptoms that mimic more common conditions like ovarian cancer or pelvic inflammatory disease. This frequent clinical overlap, combined with the lack of specific biomarkers (CA-125 being prognostic but not diagnostic), contributes to the >50% preoperative misdiagnosis rate. Nevertheless, PFTC should be seriously considered in cases of postmenopausal bleeding with negative endometrial evaluation, persistent unexplained discharge, or recurrent abnormal cervical cytology, particularly when accompanied by adnexal findings. Diagnostic difficulty underscores the need for improved imaging modalities and maintained clinical suspicion in at-risk populations [10]. PFTC should be included in the differential diagnosis with ovarian carcinoma if the patient presents with clinical symptoms such as vaginal discharge or abnormal genital bleeding or spotting with negative diagnostic curettage. Pap test positivity occurs in 10–36% of cases [11].

## 4. Pathological Diagnosis and Characteristics of Primary Fallopian Tube Carcinoma

Histopathological examination remains the gold standard for definitive diagnosis of PFTC, with high-grade serous carcinoma representing the predominant histological subtype [12]. These tumors demonstrate characteristic invasive growth patterns including papillary, glandular, and solid architectures accompanied by significant nuclear atypia, mirroring their ovarian counterparts in both morphology and biological behavior. Other histological variants, while less common, include endometrioid, undifferentiated, clear cell, mucinous, and transitional cell carcinomas, each demonstrating distinct pathological features [12]. According to Revzin et al. [13], a shared pathogenic mechanism may underline the development of peritoneal, Fallopian tube (FT), and ovarian carcinomas, with Serous Tubal Intraepithelial Carcinoma (STIC) proposed as the unifying precursor lesion. STIC, situated specifically within the fimbriated region of the FT, is posited to represent the origin of high-grade epithelial ovarian and peritoneal carcinomas (type II tumors). This lesion is recognized as the earliest histopathological stage of high-grade tubal serous carcinoma and functions as a direct precursor to invasive carcinoma of the FT. The model proposed by Revzin et al. [13] highlights STIC as a critical nexus in carcinogenesis, suggesting that malignant transformation initiates in the tubal epithelium before disseminating to secondary sites such as the ovary or peritoneum. This paradigm underscores the hypothesis that a singular pathogenic pathway, rooted in STIC, drives the progression of these anatomically distinct yet molecularly interrelated malignancies. STICs in the fimbriated end of the FT are believed to be the precursor of high-grade epithelial ovarian and peritoneal carcinomas (type II). STIC is the earliest form of tubal high-grade serous carcinoma and an immediate precursor of invasive carcinoma of the FT. These cancers are composed of noninvasive malignant tubal epithelial cells and exhibit abnormal expression of the p53 tumor suppressor protein. The precursor of STIC is a process described as a p53 signature. As STIC progresses, it may invade underlying tubal stroma or exfoliate its malignant epithelial cells onto the ovarian surface, into the peritoneal cavity, or proximally along the lumen of the FT into the uterus. This leads to the development of high-grade serous carcinoma at any one of these anatomic sites. Ovarian epithelial inclusion cysts or glands, which are probably precursors of low-grade (type I) ovarian tumors, develop by way of invaginating the ovarian surface epithelium and contain ovarian mesenchymal cells or implanted ciliated and secretory FT cells. When the ectopic glands implant onto the peritoneum or in lymph nodes, they may result in endosalpingiosis [13].

The primary tumor must originate from the tubal epithelium, demonstrate histological patterns consistent with tubal mucosa, and show clear transition zones from benign to malignant epithelium. Notably, refinements by Singh and colleagues [14] have established that even when extensive disease is present elsewhere, the identification of malignant involvement within the Fallopian tube supports a primary tubal origin, fundamentally changing traditional size-based diagnostic paradigms.

PFTC typically presents poorly differentiated lesions and exhibits aggressive biological behavior. Tumor dissemination occurs through multiple pathways including transcoelomic spread like ovarian cancer, direct extension to adjacent structures, and hematogenous/lymphatic routes. Importantly, PFTC demonstrates a greater propensity for distant metastases compared to ovarian carcinoma, though biopsy of metastatic lesions often fails to reliably distinguish between these entities [15,16]. This metastatic pattern, combined with the histological similarities to ovarian cancer, underscores the diagnostic challenges pathologists face in determining the precise origin of advanced pelvic malignancies. The evolving diagnostic criteria reflect our growing understanding of tubal carcinogenesis and have important implications for both clinical management and research classification. The diagnosis of PFTC requires the following histological features: the tumor must primarily originate from the endosalpinx, exhibit a histological pattern replicating the epithelium of the tubal mucosa, and demonstrate a discernible transition from benign to malignant tubal epithelium. Additionally, the ovaries or endometrium should either appear normal or contain a tumor smaller than the primary tubal lesion. Over 90% of PFTC cases are classified as papillary serous adenocarcinoma [12]. The identification of Fallopian tube epithelium as the tissue origin of high-grade serous carcinomas (HGSC) provides a crucial foundation for understanding the molecular mechanisms driving this aggressive malignancy. Current evidence strongly supports that the Fallopian tube epithelium represents not only the likely site of origin for most HGSCs but also an optimal model system for investigating disease pathogenesis [17].

The recognition of Fallopian tube epithelium as the progenitor tissue for HGSC has direct clinical implications, particularly for risk-reducing strategies in BRCA mutation carriers. This has led to the proposed approach of prophylactic salpingectomy with ovarian preservation, though definitive implementation requires resolution of remaining uncertainties about HGSC pathogenesis. Furthermore, the Fallopian tube epithelium offers unique opportunities for translational research, serving as an experimental platform to investigate the inflammatory and potential infectious contributors to carcinogenesis.

Table 1 summarizes the benign and malignant histotypes of Fallopian tube tumors, where possible the etiology or genetic mutation, the tissue origin of the neoplasm and were permitted in the literature the epidemiological frequency of such tumors.

## 5. From Benign to Malignant: Imaging Strategies for Accurate Fallopian Tube Pathology

PFTC cases are often misdiagnosed as ovarian carcinoma preoperatively, potentially leading to underestimation of true advanced-stage incidence [40]. Definitive diagnosis requires strict pathological criteria: the primary tumor must be located within the Fallopian tube or fimbriae, demonstrate histological transition from benign to malignant epithelium, and show either normal uterine/ovarian tissue or clearly distinct pathology. Diagnostic challenges are compounded by findings that 10–36% of PFTC patients present with adenocarcinoma-like cells on cervical cytology, while 31% show malignant findings on endometrial curettage [40,41].

Pathological conditions affecting the Fallopian tubes represent a frequent occurrence in clinical practice, ranging from entirely benign processes to malignant neoplasms.

While ultrasonography (US) and magnetic resonance imaging (MRI) serve as primary imaging modalities for FT evaluation, additional techniques including hysterosalpingography (HSG), hysterosonography, contrast-enhanced US, and computed tomography play significant roles in assessing tubal pathology. HSG is primarily used in the evaluation of infertility. However, despite the advent of new imaging modalities, HSG remains a valid procedure for visualizing the fallopian tubes, particularly in the assessment of tubal patency, as it enables the diagnosis of hydrosalpinx. Its invasive nature, and the risk of disseminating malignant cells discourage the use of this imaging method. Its diagnostic role has been entirely superseded by the superior soft-tissue characterization of MRI and the high-resolution imaging of modern ultrasound and its own technical limitations. These limitations include a lack of operator expertise, tubal muscle spasm induced by the administration of contrast medium, which can be mistaken for tubal obstruction, inadequate administration of contrast [13,14,15,16,17,18,19,20,21,22,23,24,25,26,27,28,29,30,31,32,33,34,35,36,37,38,39,40,41,42].

Contrast-enhanced ultrasound (CEUS) has solidified its role as a pivotal adjunct to conventional sonography in gynecology, distinguished by its excellent safety profile, absence of nephrotoxicity, and lack of ionizing radiation, making it ideal for patients with contraindications to other contrast agents. During a CEUS examination, the transducer is maintained in a fixed position over the region of interest. The procedure is initiated by activating a low-mechanical-index (typically <0.3) contrast-specific imaging mode to preserve the microbubble integrity. A split-screen display, juxtaposing the B-mode and contrast-enhanced views, is the preferred setup. An intravenous bolus of 1.5–2.0 mL of ultrasound contrast agent, tailored to the patient’s body habitus, is administered and followed by a saline flush. Real-time cine-loop recording is commenced upon contrast arrival to capture dynamic enhancement patterns, facilitating subsequent qualitative and quantitative analysis. Still images may be acquired for documentation and repeat injections can be performed if necessary for comprehensive lesion characterization. Its application in adnexal pathology is particularly impactful; the examination is conducted through a multiphase acquisition (arterial, late arterial, and venous phases up to 5 minutes post-injection) to meticulously analyze the dynamic wash-in and washout kinetic of the microbubble-based contrast agent within the lesion. For characterizing adnexal masses, CEUS improves the detection of neovascularization. Benign lesions, such as simple cysts or those with thin septa, typically demonstrate no enhancement or slow, progressive fill-in. In contrast, malignant lesions are characterized by rapid and high-intensity wash-in, followed by rapid washout (half-washout time > 41 s), and often exhibit irregular walls, thickened septa, and enhancing solid components or papillary projections. CEUS proves invaluable in diagnosing acute adnexal torsion by confirming the absence or reduction of parenchymal enhancement, and in evaluating tubal pathologies by demonstrating absent enhancement in torsion, mucosal hyperenhancement in pyosalpinx, or hyperenhancing masses suggesting malignancy. Limitations remain those of base ultrasound, such as operator dependence suboptimal acoustic windows, elevated body mass index, and the characteristics of the pathology itself, including deep-seated, large, or multifocal lesions [13,43]. Computed Tomography plays a crucial role in the staging of Fallopian tube carcinoma. Its primary utility lies in the delineation of metastatic disease, specifically through the identification of suspicious lymphadenopathy, peritoneal implants (carcinomatosis), and distant metastases. The presence or absence of these findings is a paramount determinant in selecting the appropriate therapeutic strategy. While CT can typically demonstrate the characteristic adnexal mass, its principal limitation is a low sensitivity for detecting early-stage disease. In these cases, high-resolution magnetic resonance imaging offers superior soft-tissue contrast and may provide critical diagnostic advantage [44,45,46].

Particularly in neoplastic lesions, advanced imaging characterization proves critical for differentiating benign from malignant processes, determining disease extent, and informing therapeutic decisions. The integration of these multimodal imaging approaches enhances diagnostic precision, underscoring their indispensable role in the comprehensive evaluation of Fallopian tube disorders [45,46,47].

### 5.1. Transvaginal Ultrasonography

Ultrasound (US) represents the first-line imaging technique for evaluating suspected Fallopian tube abnormalities and associated pelvic pathologies. Transvaginal ultrasound represents the primary imaging modality for dynamic, real-time assessment of Fallopian tube morphology and pathology.

This technique enables precise differentiation between complications including tubal torsion, hydrosalpinx, or neoplastic processes. Modern Transvaginal Ultrasound provides comprehensive morphological analysis through dimensional measurements, wall thickness evaluation, and characterization of associated abnormalities (e.g., cystic formations, solid masses, or fluid collections).

Doppler increase significantly enhances diagnostic capability by assessing vascular patterns—particularly valuable for identifying torsion-related ischemia, inflammatory hyperemia, or tumor neovascularization.

Recent technological innovations have revolutionized tubal evaluation through contrast-enhanced techniques. Hysterosonography and hybrid HSG-US modalities (utilizing B-mode and 4D imaging) employ microbubble contrast or saline infusion to precisely evaluate tubal patency

These approaches offer distinct advantages over traditional hysterosalpingography (HSG), eliminating radiation exposure risks while maintaining diagnostic accuracy for infertility assessments.

For malignant detection, prospective studies by Tongsong et al. [48] established definitive sonographic criteria for primary Fallopian tube carcinoma. The highly suggestive ‘sausage-shaped’ mass morphology—presenting as solid, solid-cystic, or cystic with intraluminal papillary projections—remains the hallmark feature.

Most sonographic findings involving the Fallopian tube demonstrate nonspecific features that overlap significantly with other pelvic pathologies, including tubo-ovarian abscess, ovarian neoplasms, and ectopic pregnancy [49]. The imaging appearance of primary Fallopian tube carcinoma varies according to its predominant morphological components—either solid tumor tissue or hydrosalpinx—with sequential imaging potentially demonstrating dynamic changes reflecting fluctuations in intraluminal serous fluid accumulation.

Two principal sonographic patterns emerge in PFTC evaluation: The first presents as a sausage-shaped adnexal mass with predominantly solid architecture, while the second manifests as a fluid-filled tubular adnexal structure containing nodular solid components or papillary projections (Figure 1 and Figure 2). An alternative presentation includes a multilocular cystic mass demonstrating characteristic cogwheel morphology on cross-sectional imaging. This variability in ultrasonographic presentation, coupled with the dynamic nature of fluid accumulation within the tubular structure, contributes to the diagnostic complexity of Fallopian tube pathology.

The morphological spectrum observed in PFTC frequently mimics both benign and malignant conditions of adjacent pelvic organs, necessitating careful correlation with clinical findings and consideration of supplemental imaging modalities for accurate differential diagnosis. The differentiation between malignant and inflammatory tubal pathology presents notable diagnostic challenges. Both pyosalpinx and acute salpingitis may demonstrate concerning features including pronounced wall thickening with folding patterns that simulate solid tumor components, as well as significant vascularity in Doppler assessment.

These overlapping characteristics necessitate careful correlation with clinical history and supplemental diagnostic testing to avoid misinterpretation. The similar sonographic presentation of inflamed mucosal folds and true neoplastic projections into the tubal lumen underscores the importance of a comprehensive diagnostic approach.

The IOTA (International Ovarian Tumor Analysis) simple rules were primarily developed to distinguish between malignant and benign ovarian masses and have been widely used in recent years (Table 2) [49,50,51,52]. According to Tongsong et al. [48], the IOTA simple rules might also be useful in differentiating Fallopian tube cancer from benign tubal disorders, as well as from ovarian cancer, particularly when combined with the recognition of Fallopian tube cancer imaging patterns. The preoperative diagnosis of Fallopian tube carcinoma was based on the IOTA (International Ovarian Tumor Analysis) simple rules, specifically the presence of M-rules (malignant features), along with at least one of the following criteria: (1) visualization of a normal ipsilateral ovary, or (2) identification of typical Fallopian tube carcinoma imaging patterns, including sausage-shaped cystic structures or lesions with incomplete septations (Figure 1 and Figure 2). A sausage-shaped cyst with a thickened wall or a cystic lesion lacking solid components or papillary projections was not considered malignant and was typically indicative of pyosalpinx or hydrosalpinx [48]. In conclusion, the sonographic features of Fallopian tube carcinoma can be summarized as follows: (1) IOTA simple rules indicating malignancy (at least one of the following: irregular solid component, presence of ascites, ≥4 papillary projections, irregular multilocular-solid tumor, or highly intense vascular flow); (2) recognition of characteristic patterns, including a sausage-shaped mass with partially or completely solid components or papillary projections, as well as lesions with incomplete septations; and (3) visualization of a normal ipsilateral ovary. During the examination, careful attention must be paid to identifying the sausage-shaped morphology (when a round or ovoid mass is observed in cross-section, the transducer should be rotated to obtain a longitudinal view to confirm its tubular configuration or incomplete septations) and meticulously document the presence of normal ovaries [48,50,51,53].

### 5.2. Magnetic Resonance Imaging

The standard diagnostic pathway for PFTC mirrors that of other gynecological malignancies, employing ultrasonography as the primary imaging modality. In cases of indeterminate or suspicious adnexal findings, magnetic resonance imaging (MRI) serves as a valuable secondary investigation [10]. For comprehensive staging evaluation, either CT or MRI may be employed, though CT’s utility remains limited to staging purposes due to its inferior soft-tissue resolution compared to MRI, rendering it suboptimal for detailed pelvic mass characterization [10]. Recent investigations by Ma et al. [54] revealed that PFTC typically presents as a small, tubular (or sausage-shaped) mass exhibiting homogeneous signal intensity—on T1-weighted imaging, the solid tumor component typically appears hypointense, on T2-weighted imaging, hypo- to intermediate signal intensity appears, which may appear either homogeneous or heterogeneous. Following gadolinium administration, the solid tumor component typically demonstrates prominent contrast enhancement, a key feature distinguishing it from the non-enhancing fluid component of associated hydrosalpinx (Figure 3 and Figure 4). Frequently associated findings included hydrosalpinx and intrauterine fluid accumulation. Ma et al. [54] identified three key diagnostic markers: (1) the distinctive tubular/sausage-shaped morphology, (2) concurrent hydrosalpinx, and (3) intrauterine fluid. These features, particularly when an adnexal mass coexists with at least one of them, significantly enhance diagnostic precision for PFTC [55]. Under normal circumstances, the Fallopian tubes are not typically visualized on routine pelvic MRI due to their small caliber and tortuous course. However, the presence of intraperitoneal fluid may render them apparent as thin, paired structures extending from the uterine cornua to the ovarian hilum within the superior aspect of the broad ligament. With an approximate anatomical length of 10–12 cm, their visualized extent often appears shorter on cross-sectional imaging due to their characteristic winding trajectory through the pelvic cavity.

The Fallopian tubes demonstrate a progressive anatomical configuration from their uterine insertion to their ovarian termination. The narrow intramural segment transitions to the isthmic portion, which subsequently widens into the ampullary region before terminating in the fimbriated infundibulum adjacent to the ovaries. This anatomical organization becomes clinically significant when pathological processes alter their normal architecture. Neoplastic transformation, particularly in cases of primary Fallopian tube carcinoma, frequently produces distinctive imaging characteristics. The tubular structure becomes distended by tumor growth, resulting in the pathognomonic sausage-shaped morphology that strongly suggests malignancy when accompanied by contrast enhancement. These lesions typically demonstrate T1 and T2 hypointensity with restricted diffusion evident on DWI sequences—features that reflect their underlying histopathological composition [10].

The radiological differentiation between malignant and inflammatory processes remains challenging due to overlapping imaging features. Both neoplastic and infectious conditions may present with luminal dilatation, solid components, and peripheral enhancement. This diagnostic ambiguity underscores the importance of correlating imaging findings with clinical history, laboratory markers, and when necessary, supplementary diagnostic procedures to establish a definitive diagnosis [56]. The variable presentation of Fallopian tube pathology, ranging from purely solid masses to complex cystic lesions with internal septations, further complicates the imaging assessment and frequently necessitates a multidisciplinary diagnostic approach [10].

Ma et al. [54], suggests that combined assessment of mass morphology (sausage-shaped configuration), enhancement patterns, and associated findings (hydrosalpinx/intrauterine fluid) improve diagnostic specificity for PFTC.

PFTC typically presents a relatively small, unilateral mass with homogeneous architecture and mild-to-moderate contrast enhancement. The characteristic sausage-shaped morphology, observed in a subset of cases, reflects tumor expansion within the Fallopian tube lumen. Hydrosalpinx, seen in approximately 30% of cases, along with intrauterine fluid accumulation due to tubal decompression, further supports the diagnosis. Mural papillary nodules projecting into the tubal lumen provide additional specificity when present [57,58]. A diagnostically significant feature is the consistent rim enhancement in PFTC demonstrates, thickness, and continuity, likely due to tumor growth along the tubular structure [59]. Quantitative analysis identifies an optimal rim thickness cutoff of 2.3 mm (AUC = 0.800) for differentiation.

Accurate identification is particularly crucial given that PFTC management requires complete surgical staging, including lymphadenectomy. In practice, radiologists should consider PFTC when encountering adnexal masses exhibiting tubular configuration, prominent rim enhancement, and secondary signs such as hydrosalpinx or intrauterine fluid. Integrating these features with clinical correlation may improve diagnostic confidence and optimize patient management. Table 3 summarizes imaging features of PFTC on magnetic resonance imaging.

#### The O-RADS MRI Score and Its Implications for PFTC Diagnosis

The Ovarian-Adnexal Reporting and Data System for MRI (O-RADS MRI) (shown in Table 4) represents a standardized, evidence-based framework for risk stratification of adnexal masses, integrating multiparametric MRI features to distinguish benign from malignant lesions [58]. Developed through recursive partitioning analysis, the system prioritizes key imaging biomarkers, including solid tissue enhancement patterns (e.g., irregular septa, papillary projections), diffusion-weighted imaging (DWI) signals, and dynamic contrast-enhanced (DCE) kinetics. Its stepwise algorithm—assessing peritoneal involvement, fatty content, and enhancement characteristics—achieves >90% diagnostic accuracy, significantly reducing false-positive rates compared to ultrasonography [59].

For PFTC, an aggressive malignancy often mimicking ovarian cancer, O-RADS MRI offers critical diagnostic advantages [60]. The system’s emphasis on detecting solid enhancing tissue within tubular structures aids in differentiating PFTC (typically O-RADS 4–5) from benign hydrosalpinx (O-RADS 2–3). Specifically, PFTC’s hallmark features—such as avid post-contrast enhancement, intermediate T2 signal, and restricted diffusion—align with O-RADS MRI’s high-risk criteria. Moreover, the score’s ability to identify peritoneal carcinomatosis (a poor prognostic indicator in PFTC) as an automatic O-RADS 5 finding underscores its utility in staging [60,61].

PFTC’s rarity has precluded robust validation within O-RADS cohorts, and its overlapping imaging features with ovarian/primary peritoneal malignancies may necessitate adjunct biomarkers (e.g., CA-125) for precise classification. Future refinements could include PFTC-specific imaging criteria, such as asymmetric tubal wall thickening or intraluminal nodularity, to enhance diagnostic specificity.

In clinical practice, O-RADS MRI streamlines triage, ensuring timely referral to oncologic centers for suspected PFTC while mitigating unnecessary interventions for benign lesions. Its integration into multidisciplinary workflows aligns with European Society of Gynaecological Oncology (ESGO) guidelines, optimizing preoperative planning and fertility-sparing decisions. Thus, while O-RADS MRI was not explicitly designed for PFTC, its systematic approach to adnexal mass characterization indirectly elevates diagnostic vigilance for this elusive malignancy. Table 4 shows the O-RADS MRI stratification system for adnexal lesions.

### 5.3. PET/CT

Hybrid positron emission tomography–computed tomography (PET-CT) represents a pivotal advancement in oncologic imaging, serving multiple critical roles in cancer management. This multimodal technique is routinely employed for primary tumor staging, therapeutic response evaluation, disease restaging, and long-term recurrence monitoring. While [^18^F] fluorodeoxyglucose (FDG) has traditionally dominated clinical PET oncology applications, it continues to maintain its status as the radiopharmaceutical cornerstone for imaging gynecologic malignancies and numerous genitourinary cancers [61]. Current evidence indicates that abdominal CT has limitations in accurately assessing the (un)resectability of advanced-stage ovarian and Fallopian tube carcinomas during primary debulking surgery [62]. Alternative imaging modalities such as PET-CT, conventional MRI, and diffusion-weighted MRI are now widely available in developed countries and may offer superior diagnostic accuracy for preoperative evaluation of macroscopic debulking (absence of macroscopically visible tumor deposits at the completion of surgery) feasibility. Among these, PET-CT demonstrates utility in tumor staging by detecting increased glucose metabolism in malignant cells, thereby improving identification of distant metastases [62]. FDG-PET/CT has become an indispensable tool in the management of gynecologic cancers, particularly epithelial ovarian and Fallopian tube carcinomas, though its application must be carefully considered within specific clinical contexts while acknowledging its inherent limitations [63]. In clinical practice, FDG-PET/CT demonstrates value in advanced-stage disease, where it surpasses conventional CT in detecting retroperitoneal lymph node involvement and distant metastases, significantly altering therapy in approximately 30% of cases [64]. Current NCCN (National Comprehensive Cancer Network) guidelines endorse its role in monitoring chemotherapeutic response and surveillance for recurrence due to its superior ability to differentiate between active disease and post-treatment fibrosis [65]. However, it is not recommended for the initial evaluation of adnexal masses, particularly in premenopausal women, due to high rates of false-positive findings related to physiological tubal or endometrial FDG uptake (related to functional cysts, endometriosis, and inflammatory processes) and false-negative results in mucinous or low-grade tumors [62].

While FDG-PET/CT shows clear superiority in identifying extraperitoneal disease, comparative studies with MRI reveal comparable performance in assessing peritoneal carcinomatosis, with MRI maintaining its status as the gold standard for local pelvic evaluation [62].

Emerging advances in radiomics and novel radiotracers (e.g., ^68^Ga-FAPI) are expanding potential applications, particularly in prognostic stratification and theranostics [66]. Nevertheless, the inherent interpretive challenges—stemming from physiological pelvic FDG avidity and overlapping with benign processes—necessitate expert integration with morphological imaging and clinical correlation [62,67]. Figure 5 demonstrates a ^18^F-fluorodeoxyglucose positron emission tomography/computed tomography findings in a case of bilateral primary Fallopian tube carcinoma with uterine metastasis. The images reveal characteristic hypermetabolic lesions involving both Fallopian tubes, along with metastatic spread to the uterine corpus, highlighting the utility of FDG-PET/CT in detecting synchronous gynecologic malignancies and mapping disease dissemination.

In conclusion, FDG-PET/CT serves as a powerful yet non-standalone modality whose clinical utility depends on judicious application in specific scenarios (advanced-stage staging, suspected recurrence) and thorough understanding of its diagnostic constraints. A multidisciplinary approach incorporating MRI and serum biomarkers remains paramount for optimal management of these malignancies [63].

## 6. The Role of Imaging in the Early Diagnosis of Primary Fallopian Tube Carcinoma

The preoperative diagnosis of PFTC remains a significant challenge, with correct identification occurring in less than 5% of cases primarily due to its nonspecific and oligosymptomatic clinical presentation [69], primarily due to its rarity and non-specific clinical presentation. However, advances in cross-sectional imaging offer critical tools that can heighten suspicion and facilitate an earlier diagnosis. This often leads to a high rate of misinterpretation, both in clinical assessment and even during surgical intervention, with literature reporting incorrect intraoperative diagnoses in 30–50% of cases [70]. Consequently, the disease is frequently identified either as an intraoperative finding or confirmed only through postoperative histopathological examination. Hysterosalpingography (HSG) demonstrates higher utility, enabling a correct or suspicious diagnosis in 82% of cases when used in conjunction with cytology. Ultrasonography, while widely used, has traditionally shown low specificity, often confusing PFTC with other tubo-ovarian pathologies (tubo-ovarian abscesses, ovarian neoplasms, or ectopic pregnancies), certain imaging features should prompt consideration of PFTC. Key characteristics include a sausage-shaped or multilobulated cystic mass, typically measuring less than 7 cm, which may demonstrate incomplete septa and highly vascularized solid components, including mural nodules or a cogwheel appearance. Advanced imaging techniques such as MRI and PET are highlighted in the literature for their superior ability to delineate tumor characteristics and metastatic spread, though data on CT remains scarce and largely uninformative [70]. The tumor marker CA-125 is deemed insufficiently sensitive for early-stage detection but proves valuable for monitoring treatment response and early identification of disease recurrence. The presence of associated findings such as hydrosalpinx, ascites, and lymphadenopathy further supports the diagnosis. Although a definitive preoperative diagnosis is uncommon, the recognition of these specific morphological patterns, particularly in a patient with clinical symptoms like abnormal vaginal discharge or bleeding and an elevated serum CA-125 level, can allow radiologists to include PFTC in the differential diagnosis of an adnexal mass, thereby guiding appropriate surgical planning and improving the potential for early intervention [71].

In summary, the preoperative diagnosis of PFTC remains complex, relying on a high index of clinical suspicion triggered by atypical presentations and the strategic integration of multiple diagnostic methods, as no single tool provides definitive preoperative confirmation.

## 7. Discussion

The routine imaging workup for any suspected gynecological malignancy typically includes US, CT, and MRI. Despite the availability of these advanced modalities, the radiological diagnosis of PFTC remains challenging.

Ultrasonography serves as an essential first-line imaging technique in the diagnostic algorithm for patients with gynecological lesions, including PFTC. Certain sonographic features, such as anechoic or low-level echoes with papillary projections or intraluminal masses, can be indicative of PFTC. However, most ultrasound appearances of the Fallopian tube are nonspecific and frequently mimic other pelvic pathologies, including tubo-ovarian abscesses, ovarian neoplasms, and ectopic pregnancies. The sonographic presentation of PFTC is largely dependent on its dominant component: the solid tumor or the accompanying hydrosalpinx. This appearance can even vary on serial imaging, reflecting changes in the volume of serous fluid within the tube. A solid-dominant tumor typically presents as a sausage-shaped adnexal mass, while a fluid-dominant presentation appears as a tubular, fluid-filled adnexal structure containing nodular or papillary solid components, or as a multiloculated cystic mass with a “cogwheel” appearance [72].

On CT, a mass lesion demonstrates attenuation similar to other pelvic soft tissues but enhances less than the myometrium [49,73]. While generally less specific, the identification of an intratubal solid papillary mass can facilitate a more confident prediction of PFTC. Regarding MRI which offers superior soft-tissue contrast, a preoperative diagnosis is more feasible when a high clinical suspicion exists for instance, in a premenopausal woman presenting with vaginal spotting and watery discharge, coupled with an elevated serum carbohydrate antigen (CA) 125 level, and is supported by characteristic MRI findings. Typically, the solid tumor component of PFTC appears homogeneously or heterogeneously isointense or hyperintense on T2-weighted images, hypointense on T1-weighted images, and demonstrates enhancement following intravenous administration of gadolinium-based contrast agents (e.g., Gd-DTPA). The hydrosalpinx component, conversely, exhibits high signal intensity on T2-weighted images and variable signal on T1-weighted images, which can be hypointense or hyperintense if hemorrhage is present [74,75]. The principal utility of MRI resides in its capacity to delineate specific signal characteristics and enhancement patterns that are pathognomonic for various diseases. For benign conditions, MRI excels in identifying the classic features of hematosalpinx, such as T1 hyperintensity from blood products and the T2 “shading” artifact, which is commonly associated with endometriosis or ectopic pregnancy. It also clearly depicts the thin, imperceptible walls and simple fluid content of a hydrosalpinx. In cases of infectious processes like pyosalpinx or tubo-ovarian abscess, MRI demonstrates thick, avidly enhancing walls, restricted diffusion, and associated inflammatory stranding within the pelvic fat.

A critical differentiating feature which differentiates malignant lesions from benign entities is the presence of enhancing solid nodular components or papillary projections within the lumen of the dilated tube. Furthermore, MRI is superior in staging disease by assessing for local invasion, lymphadenopathy, and peritoneal carcinomatosis, which are hallmarks of advanced malignancy. In conclusion, while no single imaging modality is pathognomonic, MRI emerges as the most promising tool for characterizing the intricate morphological features of PFTC. A definitive preoperative diagnosis relies on a multimodal approach that critically correlates nuanced imaging findings, particularly from MRI, with a high index of clinical suspicion.

## 8. Conclusions

Primary fallopian tube carcinoma remains a challenging diagnosis, often obscured by its similarity to ovarian cancer. This review highlights the critical role of modern imaging in improving preoperative detection. Standardized protocols like PEARLS (Pelvic Enhanced Assessment with Radiologic-Laparoscopic Staging) show promise in identifying subtle tubal lesions, facilitating earlier intervention. This review underscores the critical importance of imaging in differentiating benign from malignant lesions of the fallopian tubes. Ultrasonography serves as the primary imaging modality for the initial assessment, effectively identifying key morphological features such as tubal dilatation, wall thickening, intraluminal debris, and the presence of nodular or cystic masses. The adjunctive use of Doppler imaging further enhances this evaluation by detecting hyperemia, hyperperfusion, and neovascularization. Contrast-enhanced ultrasound offers additional valuable insight by characterizing wall enhancement patterns and the vascular architecture of solid components, thereby aiding in the preliminary diagnostic workup.

While hysterosalpingography retains a well-established role in the evaluation of tubal patency and infertility, its utility is limited in the oncological assessment of neoplastic masses.

Finally, MRI emerges as the definitive problem-solving modality due to its superior soft-tissue contrast resolution. It is indispensable for the precise local characterization of adnexal lesions and for performing local (T) staging, effectively distinguishing between benign and malignant entities. For comprehensive metastatic (M) staging, CT plays a significant role; however, its utility is surpassed by PET/CT, which provides unparalleled sensitivity for detecting distant metastases and lymph node involvement (N). It is noteworthy that CT is frequently employed as an initial emergency examination for patients presenting with non-specific symptoms, such as pelvic pain, where a broad range of differential diagnoses must be considered rapidly.

## Figures and Tables

**Figure 1 cancers-17-02985-f001:**
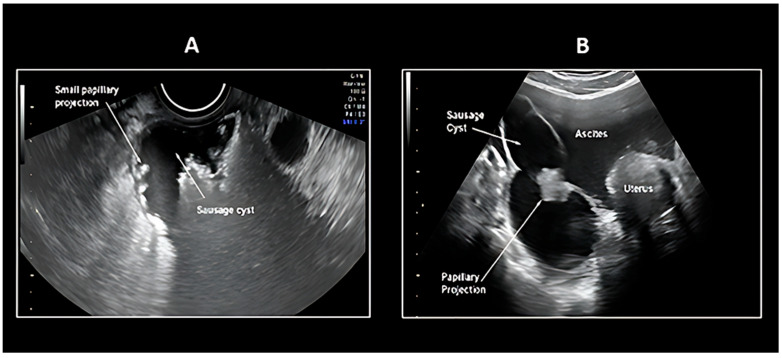
Ultrasound pattern recognition of Fallopian tube carcinoma: (**A**) Sausage cyst with Small Papillary Projections; (**B**) Sausage cyst Large Papillary Projection; [48].

**Figure 2 cancers-17-02985-f002:**
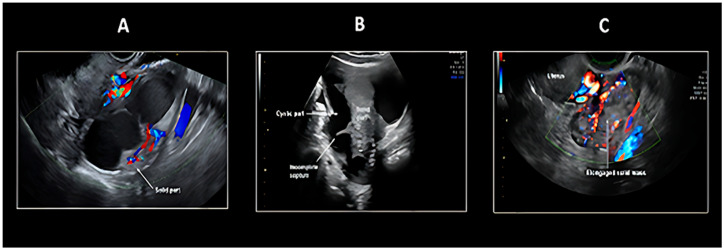
Ultrasound pattern recognition of Fallopian tube carcinoma with color Doppler assessment: (**A**) Sausage cyst with Small Area of Solid with High Vascularization; (**B**) Sausage Solid-Cystic Mass with Incomplete Septum Sausage; (**C**) Solid Mass with High Vascularization [48]. “Colours” indicate Doppler assessment.

**Figure 3 cancers-17-02985-f003:**
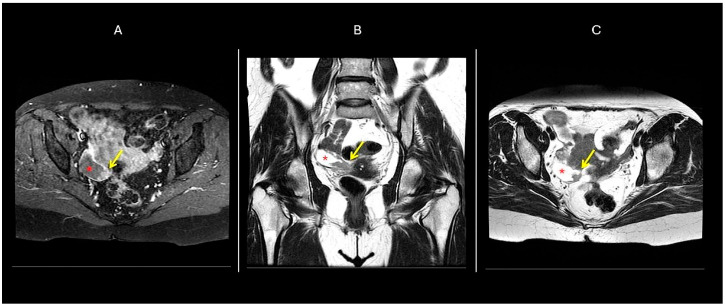
MR images demonstrate a dilated right Fallopian duct with focal mural wall thickening (arrow) exhibiting homogeneous hypeintensity on T1 fat-sat axial imaging (**A**) and hypointensity on T2-weighted coronal imaging (**B**), and on T2-weighted axial imaging (**C**). Note the associated hydrosalpinx (red asterisk) [56].

**Figure 4 cancers-17-02985-f004:**
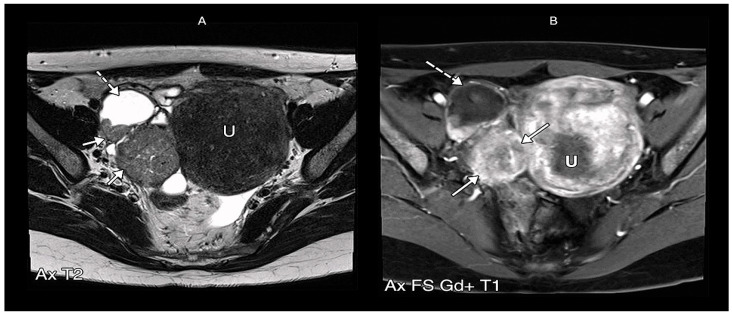
Axial pelvic MR images in a patient with Fallopian tube (FT) carcinoma. (**A**) T2-weighted image demonstrates a heterogeneous, hypointense solid mass (solid arrows) within a distended, fluid filled FT, accompanied by hydrosalpinx (dashed arrow). Small-volume pelvic ascites is noted (arrowhead). Uterus (U) is labeled for reference. (**B**) Gadolinium-enhanced T1-weighted fat-suppressed image reveals homogeneous enhancement of the uterus (U) and a similarly enhancing right FT mass (solid arrows), with persistent hydrosalpinx (dashed arrow) [13].

**Figure 5 cancers-17-02985-f005:**
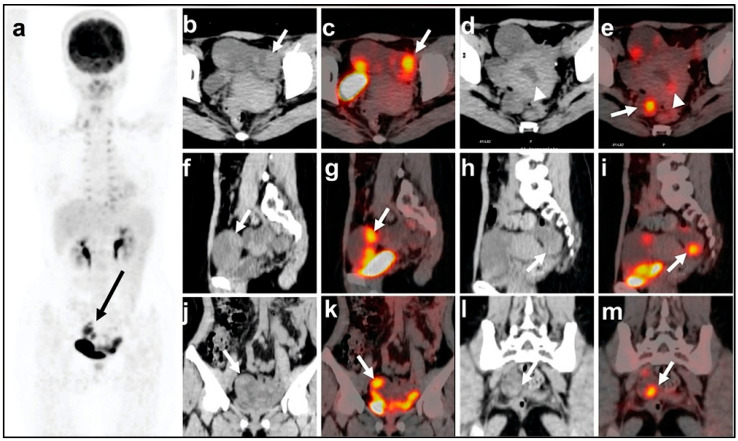
18-F fluorodeoxyglucose positron emission tomography/computed tomography findings of bilateral primary Fallopian tube carcinoma and metastasis to the uterus. A combined positron emission tomography/computed tomography (PET/CT) skull base-to-mid-thigh scan was performed 60 min after intravenous administration of 6.31 mCi of 18-F fluorodeoxyglucose (FDG). CT shows a fluid-filled tubular adnexal structure, containing nodular or papillary solid components (long arrow; left adnexal mass: transverse (**b**), sagittal (**f**), and coronal (**j**) images; right adnexal mass: transverse (**d**), sagittal (**h**), and coronal (**l**) images). Uterine lesions are not clearly shown. The maximum long diameter and vertical diameter of the mass in the axial plane are approximately 43 × 97 mm (left) and 21 × 69 mm (right), respectively. Maximum intensity projection PET (**a**) and fused PET/CT images show nodular solid components of the masses presenting with intense FDG uptake, with a maximum standardized uptake value (SUVmax) of 10.91 (long arrow indicates the left adnexal mass in transverse (**c**), sagittal (**g**), and coronal (**k**) images) and 9.28 (long arrow indicates the right adnexal mass in transverse (**e**), sagittal (**i**), and coronal (**m**) images), respectively. PET/CT also shows focal intense FDG uptake with an SUVmax of 3.55 in the uterine cavity (short arrow in (**e**)) [68].

**Table 1 cancers-17-02985-t001:** Benign and malignant histotypes of tubal tumors, the etiology or genetic mutation, the tissue origin of the neoplasm and the epidemiological frequency.

Histological Subtype	Etiology/Genetic Mutation	Origin	Frequency
Benign Neoplasms			
Papilloma	Local tubal hyperplasia in response to hormonal hyperstimulation or inflammation [18]	Epithelial Tumor	Rare
Cystadenoma	von Hippel-Lindau Disease (VHL gene mutations) [19]	Epithelial Tumor	Rare
Adenofibroma	Embryological remnant originated from the Müllerian duct [20]	Epithelial Tumor	Rare
Cystadenofibroma	Unknown [21]	Epithelial Tumor	Rare
Metaplastic papillary tumor (MPTFT)	KRAS e BRAF detected incidentally upon examination of Fallopian tube segments removed for sterilisation postpartum [22]	Epithelial Tumor	Extremely rare
Endometroid papilloma	Estrogen-driven hyperplasia [23]	Epithelial Tumor	Extremely rare
Leiomyoma	The pathogenesis of this disease is still unclear, but the nodules are thought to originate from sub-mesothelial multipotential cells located in the female pelvic peritoneum [24]	Soft Tissue Tumor	Extremely rare
Adenomatoid tumor	BRCA1 may be involved TRAF7 [25]	Mesothelial Tumor	Although Rare the Most Common among Benign Neoplasm
Mature Teratoma	Unknown [26]	Germ Cell Tumor	Extremely rare
Placental site nodule	Non involuted placental site from remote gestations in the uterus [27]	Trophoblastic Tumor	Rare
Hydatiform mole	Androgenetic diploidy (46, XX) or triploidy (69, XXY) [28]	Trophoblastic Tumor	Exceptionally rare in the Fallopian tube, usually associated with molar ectopic pregnancy
**Malignant Neoplasms**			
Serous Adenocarcinoma (Low and high grade)	TP53 (high grade) BRAF/KRAS BRCA1/BRCA2	Epithelial Tumor	The most common among malignant neoplasms (80%) [22]
Endometroid Adenocarcinoma	BRCA1/BRCA2 c-erbB-2	Epithelial Tumor	7% [22]
Mucinous Adenocarcinoma	BRCA1/BRCA2	Epithelial Tumor	2% [22]
Clear cell Adenocarcinoma	BRCA1/BRCA2 [29]	Epithelial Tumor	2% [22]
Undifferentiated Carcinoma	BRCA1/BRCA2 [30]	Epithelial Tumor	1% [22]
Transitional Cell Carcinoma	BRCA1/BRCA2 [30]	Epithelial Tumor	Rare
Squamous Cell Carcinoma	HPV-associated [31]	Epithelial Tumor	Extremely rare
Adenosarcoma	DICER1 TP53 [32]	Epithelial–Mesenchymal Tumor	Rare
Malignant Müllerian mixed tumor (Metaplastic Carcinoma, Carcinosarcoma)	Unknown [33]	Epithelial–Mesenchymal Tumor	Rare
Immature Teratoma	Unknown [34]	Germ Cell Tumor	Extremely rare
Leiomyosarcoma	Prior pelvic radiation Tamoxifen use for >5 years From a pre-existing leiomyoma Hereditary retinoblastoma and Li-Fraumeni syndrome [35]	Soft Tissue Tumor	Extremely rare
Choriocarcinoma	After complete hydatidiform mole [36]	Trophoblastic Tumor	Extremely rare
Invasive Mole	From complete hydatidiform moles [36]	Trophoblastic Tumor	Extremely rare
Placental Site Trophoblastic Tumor	Unknown [37]	Trophoblastic Tumor	Extremely rare
Lymphoma and Leukemia	Pathogen infection [38]	Lymphatic and hematopoietic Tumor	Extremely rare
Metastases	Primary tumors: Ovary, endometrium, GI tract [39]	Secondary Tumor	Variable

**Table 2 cancers-17-02985-t002:** IOTA Ten simple rules for identifying a benign or malignant Fallopian tube tumor [50].

Rules for Predicting a Malignant Tumor (M-Rules)		Rules for Predicting a Benign Tumor (B-Rules)	
M1	Irregular solid tumor	B1	Unilocular
M2	Presence of ascites	B2	Presence of solid components where the largest Solid component has a largest diameter < 7 mm
M3	At least four papillary structures	B3	Presence of acoustic shadows
M4	Irregular multilocular solid tumor with largest diameter ≥ 100 mm	B4	Smooth multilocular tumor with largest diameter < 100 mm
M5	Very strong blood flow (color score 4)	B5	No blood flow (color score 1)

**Table 3 cancers-17-02985-t003:** MRI and clinical features of PTFC.

Imaging Feature	PFTC
Morphology	Tubular/sausage-shaped mass
Laterality	Tipically, Unilateral
Size	Smaller (mean ~6 cm)
Internal Architecture	Homogeneous solid component
Enhancement Pattern	Continuous thick rim enhancement (>2.3 mm)
Hydrosalpinx	Present in ~30–50% of cases
Intrauterine Fluid	Present in ~30% of cases
T2 Signal	Hyperintense tubular structure
Diffusion Restriction	Non-specific ADC values
Doppler Flow	Moderate vascularity in solid portions
Clinical features	
Latzko triad	15% [9,10]
Hidrops tubae profluens	5% pathognomonic

**Table 4 cancers-17-02985-t004:** O-RADS MR Score [59].

O-RADS	Description
0	Incomplete exam
1. Normal ovaries	No ovarian lesion
	Physiological: Follicle Corpus luteum
2. Almost certainly benign < 0.5%	Cyst: Unilocular—simple or endometriotic fluid Thin, smooth wall with enhancement No solid tissue
	Cyst: Unilocular/multilocular—any type of fluid No wall enhancement No solid tissue
	Cyst: Unilocular/multilocular—lipid content No solid tissue
	Lesions with Solid tissue: homogeneously hypointense on T2 and DWI (dark/dark)
	Para ovarian cyst—simple fluid Thin, smooth wall with enhancement No solid tissue
	Dilated Fallopian tube simple fluid Thin, smooth wall/endosalpingeal folds No solid tissue
3. Low risk < 5%	Cyst: Unilocular—hemorrhagic mucinous or proteinaceous fluid Smooth enhancing wall No solid tissue
	Cyst: Multilocular—any type of fluid Smooth septae and enhanced wall No solid tissue
	Lesion with solid tissue (excluding T2 dark/DWI dark) Low risk time intensity curve on DCE MRI
	Dilated Fallopian tube—non-simple fluid No solid tissue
4. Intermediate risk 5–90%	Lesion with solid tissue (excluding T2 dark/DWI dark) Intermediate time intensity curve on DCE MRI Enhancing < or = myometrium at 30–40 s on non-DCE MRI if TIC unavailable
	Lesion with lipid content Large volume solid tissue
5. High risk > 90%	Lesion with solid tissue High risk time intensity curve on DCE MRI Enhancing > myometrium at 30–40 s on non-DCE MRI
	Obvious peritoneal, mesenteric, or omental nodularity or thickening

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
