# Peer review of "The Imaging of Primary Fallopian Tube Carcinoma: A Literature Review"

_cancers, 2025, doi:10.3390/cancers17182985_

Round 1
Reviewer 1 Report
Comments and Suggestions for Authors
Comments to the article “Primary Fallopian Tube Carcinomas: An Underestimated Entity Masquerading as Ovarian Cancer – The Role of Imaging and Comprehensive Diagnostic Approaches” submitted by Iacobellis et al. to Cancers.
Primary Fallopian Tube Carcinomas are rare and limited clinical cases and reports have been published. Among the gynecological malignancies, PFTC form a small group and the expertise to diagnose it expectedly limited. Using imaging tools for diagnosis of fallopian tubes and related malignancies have been reviewed earlier (Berek et al., 2015 DOI: 10.22034/APJCP.2017.18.11.3011; Revzin et al., 2020 DOI: 10.1148/rg.2020200051; Tonsong et al., 2017 DOI: 10.22034/APJCP.2017.18.11.3011). Recent reviews have emphasized on imaging of FTs (doi: 10.3390/cancers16081560, DOI: 10.1002/jum.16628, DOI: 10.1016/j.mric.2022.06.008 ). Within this domain, the authors have selected a niche area of PFTC for literature review-based analyses.
The authors must revise their manuscript as per given comments.
- Section 2. The approach of literature collection and its evaluation are poorly thought of. It is now widely accepted that a literature survey-based report is considered as a Systematic Review and an analysis of the data generated through the systematic review is referred to as Systematic Review and Meta-Analyses. The authors should refer to the PRISMA guidelines (https://www.goodreports.org/reporting-checklists/prisma/) for the same.
- Section 3 onward the manuscript takes a shift from a systematic approach to literature review approach and present the already reviewed work like the Section 4 (Clinical Features), and Section 5 (Pathological diagnosis and characteristics of primary fallopian tube carcinoma). Section 6 is crucial and becomes part of the existing discussion regarding the origin of HGSOCs. However, this section is very short and the Table 2 adds limited value. Section 7 on biomarkers is misplaced in here. It seems that the authors have tried to forcefully merge two different reviews – systematic review-meta-analyses and review. But they failed to present a single picture. Considering the aim of the review has been on the imaging for diagnosis of PFTCs, sections 3-7 may be removed altogether.
- Section 8 onward is the centre of the proposed manuscript. Here the authors have referred to imaging techniques and used select case reports to make their point.
- Lines 263-276 is a repeat of the introduction and is not required. Such details are preferred for a Book / Book Chapter over the Review.
- Line 289-309 is an introduction to transvaginal ultrasonography method. It may be reduced to give space to review of the clinical cases of PFTCs.
- Section 8.3. A background of the method and its advantages are elaborated. However, cases of PFTC are expected to be reviewed and discussed here.
- Section 9. This section is a distraction the focus of the manuscript. It is irrelevant to the current review and should be removed from the manuscript.
- Section 10. Discussion. This section is poorly written and appears as an extension of the introduction. The authors have reviewed the PFTC but have not mentioned a single sentence of the use of imaging techniques and how they may aid in diagnosis of PFTCs. This section is expected to have detailed comparison of various imaging methods. As such this section may be dropped from the manuscript.
Author Response
We sincerely thank the Reviewer for their thoughtful evaluation of our manuscript, and we have made changes as suggested to make the text more usable
1)Section 2: The present study constitutes a narrative literature review rather than a systematic review or meta-analysis. Consequently, strict adherence to the PRISMA (Preferred Reporting Items for Systematic Reviews and Meta-Analyses) guidelines is not required, as these are specifically intended for systematic reviews that employ rigorous methodological protocols, predefined search strategies, and quantitative data synthesis. This article adopts a traditional review approach, which permits a more comprehensive and interpretative examination of the existing literature without the formal procedural constraints of a systematic review. Therefore, while the PRISMA guidelines serve as a robust framework for systematic
reviews, their application is not obligatory for narrative reviews of this nature.
2) We have expanded the section on ‘Differences and Similarities Between Primary Fallopian Tube Cancer and Ovarian Cancer’ and Table 2 was delated. Current pathological classifications consider PFTC and EOC, particularly high-grade serous ovarian carcinoma, as a single nosological entity based on their overlapping histopathological and biomolecular profiles. However, accumulating clinical evidence demonstrates that these malignancies exhibit distinct patterns of lymph node metastasis, clinical presentation, and imaging characteristics (Y. Yang et al. / Clinical Radiology 75 (2020) 457e465; Alvarado-Cabrero et al., 2013). Comparative studies have consistently shown that PFTC displays a greater propensity for lymphatic spread, with significantly higher rates of retroperitoneal and para-aortic lymph node involvement compared to EOC. These differences in metastatic behavior, along with variations in clinical manifestations and radiological findings, persist despite their shared molecular pathogenesis.
3) Lines 263 and 276 were deleted.
4) Line 289-309 were reduced.
5) In section 8.3 that has become 6.3, considering the paragraphs that it was recommended to delete (section 3 and 7), it introduces the PET/CT technique applied to PFTC. A case of bilateral PFTC is presented, explained in the text and caption in Figure 4.
6) Section 9 was deleted.
7)The Discussion section was reworded again, emphasising the crucial aspects of PFTC, a truly rare gynaecological oncological disease, and emphasising the value of imaging for diagnosis, highlighting how the latter can differentiate ovarian from tubal cancer
Reviewer 2 Report
Comments and Suggestions for Authors
The manuscript is a review article focused on rare gynecological malignancies, specifically Primary Fallopian Tube Carcinomas (PFTC). It provides a detailed overview of symptoms, clinical features, and histopathological examination of PFTC. Special attention is given to the challenges involved in providing an accurate diagnosis of this condition. More specifically, the authors compare the differences and similarities between PFTC and ovarian cancer. Additionally, a significant part of this review discusses imaging modalities used to observe PFTC, including Transvaginal Ultrasonography, magnetic resonance imaging, and hybrid PET/CT. These sections constitute the core content of the manuscript. To some extent, the authors also consider strategies for treatment and therapy, offering suggestions to improve current procedures, particularly regarding the potential of imaging modalities.
The paper is well written and organized. Readers can easily follow the main ideas reflecting the current state of knowledge in this area of clinical medicine. The references cited thoroughly cover the topic, including the existing gaps, challenges, and potential solutions. The explanations are clear and evidence-based. Furthermore, unlike many modern publications, the authors have made an effort to edit the text to a high standard. This manuscript deserves consideration for publication as a review article in the journal.
I recommend this article for publication in Cancers.
Author Response
We sincerely thank the Reviewer for their thoughtful evaluation of our manuscript and for recognizing the efforts we have made to comprehensively address this challenging topic in gynecological oncology. We are particularly grateful for the positive assessment of our work's organization, clarity, and editorial quality. The Reviewer's insightful comments regarding the thorough coverage of diagnostic challenges, comparative analysis between PFTC and ovarian cancer, and detailed examination of imaging modalities are especially appreciated. We are encouraged by the recognition of our evidence-based approach and the balanced discussion of current limitations and potential improvements in diagnostic and therapeutic
strategies. Thank you for considering our manuscript worthy of publication. We believe this review will contribute meaningfully to the literature on rare gynecological malignancies and help improve the diagnosis and management of PFTC in clinical practice.
Reviewer 3 Report
Comments and Suggestions for Authors
Fallopian Tube Cancer
The goals of this manuscript are this manuscript are unclear. It spends a lot of time trying to differentiate ovarian cancer from fallopian tube cancer, and that is no longer considered necessary. The majority of experts just combine the high-grade serous cancers (HGSC) into tubo-ovarian cancers to describe the likely origin of the malignant cells from the fallopian tube and then the remainer of the malignancy is likely metastatic tumor. Early tubal lesions found incidentally at prophylactic salpingectomy or at the time of other surgery are usually the only primary HGSC of the fallopian tube. This is unlikely to be known prior to the pathologic examination and we have failed to preoperatively diagnose these lesions after decades of trying.
Understanding the P53 mutation in HGSC, and the common genetic mutations associated with these cancers is important, so that proper family counseling, and cascading of genetic testing and preventative surgery, and screening can occur in family members. The need for all pathologists to completely submit all fallopian tubes removed to thorough histologic examination called SEE-FIM (sectioning and extensive examination of the fimbraed end) will be the only way to diagnose these lesions accurately. The American College of Pathology still has not dictated universal SEE-FIM for all tubes removed.
If the purpose of this manuscript is to describe all possible lesions of the fallopian tube then table 1 makes some sense, except it does not appear to be accurate. Leiomyoma is miss spelled, metplastic (metaplastic?), TP53 is mainly associated with high grade serous carcinoma (not low grade or endometrioid or mucinous), BRCA 1&2 is mainly HGSC rather than all epithelial cancers.
Table 2 is not helpful, since it is best to combine ovary and fallopian tube cancers that are HGSC and not important to differentiate them. All HGSC need genetic testing and should be TP53 positive and are treated the same as tubo-ovarian cancer.
The attempt to have radiology make a pathologic diagnosis in section 8 seems unrealistic. Wasting time and money on PET, MRI, CT, ultrasound, when operative diagnosis is necessary. Knowing when to refer to a gynecologic oncologist vs a gynecologist is the only strategy that makes sense unless the patient is trying to preserve their fertility.
The section 9 and table 6 seems beyond the scope of this manuscript and likely will be out of date by the time the paper is published. The purpose of the manuscript needs to be narrowed or the emphasis should change. It is not realistic to separate HGSC into tubal vs ovarian, and is no longer considered relevant. To much of this manuscript is wasted on these thoughts. I reject the discussion and conclusions that it is important to differentiate tubal origin vs ovarian when histology is HGSC. It may be best for the authors to concentrate and focus on lesions of the fallopian tube that are not HGSC?
Author Response
We sincerely appreciate the reviewer's valuable insights regarding the evolving classification of tubo-ovarian HGSC. We fully acknowledge the current consensus that these malignancies represent a single nosological entity from a histomolecular perspective, as clearly stated in our manuscript. However, the primary objective of this literature review was to specifically evaluate the role of imaging in diagnosing primary fallopian tube carcinoma (PFTC), particularly focusing on how these tumours are frequently underdiagnosed as ovarian malignancies. Our analysis revealed that, despite their molecular similarities, certain semiological criteria in the literature do permit radiological differentiation between these entities. Current pathological classifications consider PFTC and EOC, particularly high-grade serous ovarian carcinoma, as a single nosological entity based on their overlapping histopathological and biomolecular profiles. However, accumulating clinical evidence demonstrates that these malignancies exhibit distinct patterns of lymph node metastasis, clinical presentation, and imaging characteristics (Y. Yang et al. / Clinical Radiology 75 (2020) 457e465; Alvarado-Cabrero et al., 2013). Comparative studies have consistently shown that PFTC displays a greater propensity for lymphatic spread, with significantly higher rates of retroperitoneal and para-aortic lymph node involvement compared to EOC. These differences in metastatic behavior, along with variations in clinical manifestations and radiological findings, persist despite their shared molecular pathogenesis. We recognise the diagnostic challenges highlighted by the reviewer - indeed, our manuscript acknowledges that incidental discovery of early tubal lesions remains
the exception rather than the rule. Nevertheless, given the paucity of comprehensive reviews on PFTC imaging characteristics and the rarity of this malignancy, we believed a thorough examination of the existing literature could yield valuable insights for radiologists and clinicians encountering these challenging cases.
We appreciate the reviewer's expertise in identifying these important issues. The revised table now provides a more precise and evidence-based overview of fallopian tube lesions and their molecular characteristics. These changes have strengthened the manuscript's accuracy and clinical utility. The updated Table 1 appears of the revised manuscript, with all modifications clearly highlighted for ease of review.
Based on above, table 2 was delated.
While pathological confirmation remains the gold standard, this literature review highlights how imaging can significantly impact diagnosis, prognosis, and follow-up by identifying radiological criteria that help differentiate tubal from ovarian malignancies.
Transvaginal ultrasound, as the first-line modality, plays a crucial role in initial risk stratification by distinguishing benign from suspicious adnexal masses. When findings are equivocal, MRI provides superior soft-tissue characterization, improving diagnostic accuracy for tubal origin. PET/CT, though limited to staging, aids in detecting occult metastatic disease, particularly lymph node involvement, which
influences surgical planning. We have refined the manuscript to clarify that imaging serves as an adjunct to pathology, optimizing triage and therapeutic strategies while emphasizing its limitations in replacing tissue confirmation.
The revised manuscript now more clearly emphasizes that the primary purpose of this literature review was to evaluate imaging characteristics of primary fallopian tube lesions. We have removed the extensive comparisons between tubal and ovarian HGSC that were deemed beyond the scope, while retaining only those imaging features that may suggest tubal origin when present.
Regarding the focus on HGSC, we recognize this was largely due to the extreme rarity of other histological subtypes in the fallopian tube, with most non-HGSC cases reported only as isolated case reports in the literature. The paucity of robust imaging data on these rare variants limited our ability to provide meaningful analysis.
Round 2
Reviewer 3 Report
Comments and Suggestions for Authors
New reference "current medical imaging": 2025 vol 21 page 1-7 Tian, Tongtong The Typical Computed Tomography Findings of Primary Fallopian Tube carcinoma.
The author is failing to accept the fact that most experts in both fallopian tube cancer, ovarian and primary peritoneal high grade serous cancers are all one entity in a continuum. I praise the authors on trying to help clinicians not to forget that fallopian tube cancer can on a rare occasion be diagnosed as stage one. Those patients are often BRCA 1&2 carriers undergoing a risk reducing surgery. But, we can help educate about the rare case where bleeding is occuring with a dilated tube could be an early cancer with a negative work up to include evaluation of the cervix and endometrium. I would appreciate them to decrease the comparison between high grade serous ovarian cancer and high grade serous fallopian tube cancer since that is not helpful. They are one and the same when found as early stage and peritoneal carcinomatosis of high grade serous is often after a TAH BSO when the fallopian tube was not examine using SEE-FIM.
The authors need to improve their goal of helping clinicans find stage I fallopian tube cancers and stop trying to separate the high grade serous diagnoses. There are other lesions in the fallopian tube that can remain emphasized. I am not a radiologist- and this seems like a radiology focused paper, but misleading - since all high grade serous is now considered tubo-ovarian high grade serous.
Use the papers updating this continuum and their goals of early detection of fallopian tube cancer - before it has spread to the ovary and omentum/peritoneum
This paper is very repetative and could be shortened if they would stop saying the same thing over and over (which is unnecessary) .
It would be nice to just lead with the goal of helping clinicians diagnose a very rare entity of stage I fallopian tube cancer, preoperatively, where vaginal bleeding or discharge and a dilated fallopian tube are found. The patient will have negative evaluation of the cervix and endometrium. It is not important to spend so many pages trying to compare ovarian high grade serous from fallopian tube cancer. You are just trying to find fallopian tube cancer as stage I before it metastasizes to the ovary or peritoneum. That would make more sense
Author Response
We agree entirely with the reviewer on this critical point. Upon careful reconsideration, we recognize that the extensive comparative elements between ovarian and fallopian tube HGSC were outdated and did not serve the paper's ultimate goal. Consequently, we have removed the entire section dedicated to this direct comparison. In its place, we have reframed the narrative to emphasize that from a radiological perspective, the initial imaging finding of a mass may be localized to the adnexa, and the specific goal of this paper is to aid in the early identification of a mass confined to the fallopian tube before metastatic spread, which represents a critical window for intervention. This shift in focus aligns with the concept of a tubo-ovarian continuum while highlighting the specific scenario where a radiological diagnosis can have a profound impact on prognosis. A new dedicated section has been added entitled "Imaging Features of Early-Stage Fallopian Tube Carcinoma" which details the key clinical presentation (including vaginal bleeding or discharge) and the corresponding imaging findings on MRI (specifically, a hydrosalpinx-like dilated tube with internal solid components or nodularity), always within the context of a negative workup for other causes.
We have consulted the suggested paper and others on the topic. While we agree that CT is paramount for staging disease extent, we have chosen to maintain our focus on MRI as the primary modality for characterizing primary adnexal masses and achieving the early, pre-operative diagnosis that is the core of our revised manuscript. This is due to MRI's superior soft-tissue contrast resolution, which is essential for delineating intricate pelvic anatomy and characterizing the internal architecture of a potentially malignant fallopian tube.
Round 3
Reviewer 3 Report
Comments and Suggestions for Authors
This update is better, but still could be improved by focusing on earlier diagnosis of early fallopian tube cancer. The first two paragraphs were not improved, but there is adequate improvement of the body of the manuscript for publication.
I recommend that the beginning of the paper (summary and abstract) that the emphasize the improved diagnosis of stage I fallopian tube cancer
Author Response
Thank you for your insightful comments and constructive feedback. In direct response to your recommendation, we have revised the manuscript to place a stronger emphasis on the critical importance of early diagnosis. The Simple Summary, Abstract, and Introduction have been reframed to more explicitly highlight the role of advanced imaging techniques in improving the preoperative detection of primary fallopian tube carcinoma.